# The Distribution and Predictive Factor of Extra-Pancreatic Malignancy Occurrence in Patients with Pancreatic Intraductal Papillary Mucinous Neoplasm—A Ten-Year Follow-Up Case–Control Study in Taiwan

**DOI:** 10.3390/cancers16234102

**Published:** 2024-12-07

**Authors:** Sheng-Fu Wang, Chi-Huan Wu, Kai-Feng Sung, Yung-Kuan Tsou, Cheng-Hui Lin, Mu-Hsien Lee, Nai-Jen Liu

**Affiliations:** 1Department of Gastroenterology and Hepatology, Chang-Gung Memorial Hospital, Linkou Medical Center, Taoyuan 333, Taiwan; shanelily@msn.com (S.-F.W.); b9002076@adm.cgmh.org.tw (C.-H.W.); flying@cgmh.org.tw (Y.-K.T.); linchehui@adm.cgmh.org.tw (C.-H.L.); r5266@adm.cgmh.org.tw (M.-H.L.); milk1372@cgmh.org.tw (N.-J.L.); 2School of Medicine, College of Medicine, Chang-Gung University, Taoyuan 333, Taiwan

**Keywords:** extra-pancreatic malignancy, intraductal papillary mucinous neoplasm, cyst size

## Abstract

Although extra-pancreatic malignancies may be more likely to occur in patients with pancreatic intraductal papillary mucinous neoplasm, no definite risk factor has been found to date. We aimed to identify such a risk factor in this study in order to further determine who needs cancer screening in patients with diagnosed intraductal papillary mucinous neoplasm. Additionally, we present the distribution of extra-pancreatic malignancies in Taiwan with extended follow-up periods, recommending which cancers are most prevalent and require more attention.

## 1. Introduction

Pancreatic intraductal papillary mucinous neoplasm (IPMN) is considered a precursor to pancreatic cancer. However, the malignant potential differs based on the subclassification of IPMN. One study showed that the five-year actuarial risk of pancreatic malignancy was 15% for branch duct type IPMN (BD-IPMN) and 63% for main duct type IPMN (MD-IPMN) (*p* < 0.001) [1]. To date, the strategy for managing these patients and making decisions regarding operation is based on the Fukuoka criteria, with the main goal being the prevention of pancreatic cancer [2].

Multiple studies have demonstrated an increased incidence of extra-pancreatic malignancies (EPMs), with prevalence rates of 10–52% in patients with pancreatic IPMNs. Among these, gastrointestinal tract cancers are the most commonly observed, with colon cancer being more prevalent in Western countries and gastric cancer more common in Asia [3,4]. One case–control study assessed 178 patients with resected IPMN, comparing them to 356 patients in an age- and gender-matched control group. Their data showed that EPMs were more common in patients with IPMN than in the control group (16.8% vs. 8.4%, *p* = 0.003). In addition, most of their EPMs (70%) occurred preceding IPMN diagnosis [5]. Besides the evidence that EPMs are more frequent and mostly precede or occur concurrently with the diagnosis of IPMN, a population-based study in 2019 found that patients diagnosed with malignant IPMN also had an increased risk of developing an EPM. This study assessed 2850 patients who showed a more advanced stage of IPMN with a shorter period from the diagnosis of IPMN to an EPM diagnosis and a higher risk of EPM occurrence [6].

After reviewing previous reports, we found that most EPMs were diagnosed before or concurrently with the diagnosis of IPMN, with the incidence ranging from 66% to 94% [7,8,9,10]. It is supposed that IPMN is usually an accidental finding during routine image examination for patients who have already had other malignancies. Another issue is follow-up duration. The duration of follow-up ranged from 14 to 50 months in previous studies. Thus, we planned to investigate whether EPMs are also an important issue to be monitored after IPMN diagnosis and when they typically occur if the follow-up duration is extended [3,4].

Although many studies have proved the correlation between IPMN and EPMs, a few risk factors were found to be associated with those diagnosed with an EPM before or concurrently with IPMN diagnosis, as well as the prediction of EPMs after IPMN diagnosis. A previous study assessed 61 patients with a pathological diagnosis of IPMN and found that EPMs occurred more often in patients with benign IPMN than malignant IPMN (N = 10/25 vs. 5/36, *p* = 0.033) [8]. In contrast, only malignant IPMNs had a significant association with prior EPMs in another retrospective study, showing a near four-fold increase in incidence compared to the general Japanese population [11]. Although old age was found to be a risk factor of EPMs in a previous study in Korea, another study revealed that clinical and pathological features including age, sex, family history of malignancy, history of cigarette smoking, alcohol abuse, and the type of IPMN were not different between patients with and without an EPM [10,12]. The expression of MUC2 (*p* = 0.04) in surgical specimens seems to be more frequent in IPMNs coexisting with gastrointestinal cancer compared to p21 (*p* = 0.12), p53 (*p* = 0.25), and MUC5AC (*p* = 1.0) [13]. However, this biomarker can only be examined when patients undergo surgery for IPMNs, and most IPMNs are small in size and only need image surveillance. At present, there are no practical tools or clinical biomarkers available for identifying groups at high risk of EPM occurrence after IPMN diagnosis. Consequently, our aim was to assess these patients and document their cyst features using MRI and CT scans in order to identify predictive risk factors for EPM occurrence.

## 2. Materials and Methods

We retrospectively collected data from 196 patients with IPMN who were diagnosed via magnetic resonance imaging (MRI) from 2010 to 2014 in Chang Gung Memorial Hospital. All of the images were reviewed by our radiologist with expertise. After excluding those patients with a follow-up duration of less than one year and/or those whose pathology showed a diagnosis other than IPMN in patients having surgery, 114 patients were enrolled for analysis. Among these patients, 95 patients had a serial image follow-up for features of cysts, and the other 19 patients initially underwent surgery with high-risk stigmata when diagnosed (Figure 1).

The radiologic criteria for diagnosing MD-IPMN included the presence of communication with the main pancreatic duct, accompanied by dilatation. For BD-IPMN, the criteria included dilated branch pancreatic ducts presenting as a cluster of small cysts with a grape-like appearance, a multilocular cyst with papillary projections, or a single cystic lesion with a lobulated or irregular margin that communicates with the main pancreatic duct [14].

The characteristics collected from these patients included age, gender, IPMN type differentiated by MRI (main duct, branch duct, or mixed), cyst location (uncinate process, head, neck, body, or tail), number of cysts, size of the cyst when initially diagnosed and at last follow-up date, common bile duct (CBD) diameter, and main pancreatic duct (MPD) diameter. For patients with a normal p-duct size, the p-duct size at the pancreatic body was measured, and for those with a focal dilated or irregular p-duct size, the maximal p-duct size was measured. Other recorded parameters included whether a mural nodule was noted under MRI; duration of follow-up; and biochemical data including glycohemoglobin (HbA1C), liver biochemistry, carcinoembryonic antigen (CEA), and carbohydrate antigen 19-9 (CA 19-9). The types and timing of EPM occurrence were recorded as well.

The maximum cyst size was measured in the coronal plane and assessed at the initial and final cross-sectional imaging studies. The surveillance interval was defined as the time between these two scans and varied for each patient. In cases of multifocality, the size of the largest cyst was considered. The growth rate was calculated as the change in cyst size over the number of surveillance years.

We used univariate and multivariate logistic regression tests to assess the odds ratio of possible risk factors of EPMs, including age, gender, and biochemical data such as CEA, CA-199, neutrophil–lymphocyte ratio (NLR), platelet–lymphocyte ratio (PLR), and HbA1C. The cyst features were analyzed as well, including type of IPMN, location of the IPMN, whether the cyst is multifocal, the number of cysts, the size of the cyst when diagnosed, whether the mural nodule noted by MRI, and the cyst progress rate, which was calculated as the size change from the initial diagnosis to the last image divided by follow-up years. We also used univariate and multivariate logistic regression tests to assess the odds ratio of potential risk factors of EPMs only occurring after IPMN diagnosis. Further, we used the AUROC curve and Youden’s index to find a cut-off value if a predictor was found with statistical significance that could be used for clinical practice.

Numerical data are presented as means with standard deviations when they conform to a normal distribution, and as medians with quartiles when they do not. Categorical data are shown as frequencies and percentages. We used SAS 9.4 software to perform the statistical analyses. An independent *t*-test was used to compare normally distributed continuous variables, while the Mann–Whitney U test was applied to non-normally distributed variables. Categorical variables were compared using a Chi-square test. When more than 20% of the expected frequencies were less than 5, Fisher’s exact test was used. *p*-values of less than 0.05 were considered statistically significant.

## 3. Results

In total, we assessed 114 patients diagnosed with IPMN by MRI, with an average follow-up period of 10.45 years. Among these patients, 42 (36.8%) experienced EPM events, including those that occurred before, concurrently with, or after the diagnosis of IPMN. The number of male and female patients was equal, without statistical significance between the groups with and without EPM occurrence (*p* = 0.114). Sixty-two patients (54.4%) were over 65 years old, and the group with EPM occurrence had a higher proportion of older patients (N = 28 vs. 34, 66.7% vs. 47.2%, *p* = 0.02). Regarding the cyst features, most patients had branch type IPMN (N = 102, 89.5%) and only two patients had mixed type IPMN. Furthermore, 58 patients (50.9%) had cysts located in the pancreatic body, with the second most common location being the uncinate process (N = 39, 34.2%). Additionally, the median size of the cyst when initially diagnosed was 1.5 cm, and most patients had less than three cysts (N = 107, 84.2%). Nineteen patients (16.7%) had a mural nodule shown by MRI initially, and the EPM group had more patients with mural nodules (N = 11 vs. 8, 26.2% vs. 11.1%, *p* = 0.025). The average biochemical data were as follows: HbA1C, 6.55 (5.7–7.35); aspartate aminotransferase (AST), 20 (16–25); alanine aminotransferase (ALT), 19 (15–25); CEA, 1.75 (1.18–3.22); CA-199, 14.59 (6.92–30.00). The characteristics of these patients are shown in Table 1.

The indications for the original imaging which detected the pancreas cyst are shown in the Appendix A. The main indication for performing CT/MRI was abdominal pain (N = 54/114, 47.3%), although IPMN was usually an incidental finding and may not have been the leading cause of the pain. The second most common indication was EPM staging or follow-up (N = 21/114, 18.4%), followed by IPMN as an incidental finding during a health examination (N = 12/114, 10.5%). Additionally, we recorded the type of malignancy in patients with an EPM and further classified it as occurring before, concurrently with, or after the diagnosis of IPMN, as shown as Table 2. Several types of malignancies were found, with the most common being colon cancer (N = 10, 21.3%) and lung cancer (N = 10, 21.3%), followed by hepatocellular carcinoma (N = 9, 19.1%) and urothelial cancer (N = 5, 10.6%). Notably, most IPMN patients developed lung cancer after IPMN diagnosis (N = 8/10, 80%). According to the medical records, a total of 47 malignancies appeared in 42 patients. Overall, 21 malignancies (44.7%) were found before or concurrently with IPMN diagnosis, and 26 malignancies (55.3%) were found after IPMN diagnosis when the follow-up period was extended up to 10.45 years. We also compared the incidence of cancers in Taiwan with the incidence of EPMs after an IPMN diagnosis, using the recent statistical data of the Ministry of Health and Welfare in Taiwan. It is worth mentioning that only 6 out of 26 patients (23.1%) were diagnosed with an EPM during the regular surveillance for IPMN follow-up. This highlights the importance of maintaining extra vigilance for EPM occurrence and the need for other screening methods after IPMN diagnosis. The mean duration from IPMN diagnosis to EPM occurrence was 7.4 ± 3.7 years, and the median age at EPM diagnosis was 75.6 (65.2–81.2) years.

The risk factors of EPM occurrence in IPMN patients with serial image follow-up were analyzed using univariate and multivariate logistic regression tests, with the results shown in Table 3. There was no statistical significance of age and gender. The cyst features demonstrated that the increase in cyst size (*p* = 0.000, OR = 9.429, 95% CI = 3.565–24.942) and the rate of cyst progression (*p* = 0.004, OR = 188.399, 95% CI = 5.08–999) were statistically significant. Of the biochemical data, only CEA had statistical significance (*p* = 0.030, OR = 1.366, 95% CI = 1.030–1.811), whereas NLR, PLR, and HbA1C did not show significance under the univariate logistic regression test. Further, only the increase in cyst size (*p* = 0.004, OR = 8.542, 95% CI = 1.979–36.862) had statistical significance under the multivariate logistic regression test, and the AUROC curve showed an area under the curve of 0.900 (Figure 2).

Moreover, we sought to identify the predictors of EPM occurrence after IPMN diagnosis using the same factors, and the results are given in Table 4. It appeared that only the increase in cyst size had statistical significance under the univariate logistic regression test (*p* = 0.001, OR = 3.026, 95% CI = 1.538–5.952) and the multivariate logistic regression test (*p* = 0.002, OR = 2.911, 95% CI = 1.446–5.861). The AUROC curve showed an area under the curve of 0.8102 (Figure 3). We determined that a cut-off value of 1 cm had clinical utility (accuracy, 79%; sensitivity, 88%; specificity, 58%).

## 4. Discussion

Our results revealed that 42 patients (36.8%) had EPM occurrence, consistent with previous reports. Notably, 55.3% of these EPM occurrences happened after IPMN diagnosis when the follow-up duration was extended to 10.45 years. That is significantly higher than previous studies, which showed that most EPMs occurred before or concurrently with IPMN diagnosis, even in one prospective study in Japan [3,15,16]. The rate of EPM occurrence after IPMN diagnosis ranges from 4% to 15% for all EPMs detected in different reports [4,10,12,17]. This difference is likely due to the follow-up duration, as most studies averaged 14–50 months [16,17,18], whereas the mean duration from IPMN diagnosis to EPM occurrence in our study was 7.4 ± 3.7 years. In addition, patients with IPMN in previous studies were diagnosed with a resected specimen because of high-risk stigmata or worrisome features of IPMN. Riall and colleagues detected EPMs in 86% of cases before and only in 14% of cases after IPMN diagnosis, with their population consisting of only 5% benign IPMNs [19]. These patients rarely survived long enough to develop other malignancies [5,7,11,20]. Although our study only included 114 patients, it is the first to follow a group of non-invasive IPMN cases for more than 10 years (median: 10.45 years) and has demonstrated the distribution of EPM occurrence after IPMN diagnosis in the literature, to the best of our knowledge.

EPM prevalence reflects the geographic distribution of malignancies in different countries. Previous studies have shown that gastric cancer is the most frequent EPM in Eastern populations, while colorectal cancer is the most common in Western patients. This may be explained by the similar sequence from adenoma to adenocarcinoma [21,22]. In our study, lung cancer and colon cancer were the most prevalent EPMs, with hepatocellular carcinoma being the second most common. Notably, most lung cancers occurred after IPMN diagnosis (N = 8/10, 80%), whereas only three patients (30%) developed colon cancer after IPMN diagnosis. Lung cancer as an EPM in IPMN patients has also been mentioned in two previous studies. Eguchi et al. revealed that colon cancer was the most common EPM (N = 8/69, 12%), followed by lung cancer (N = 5/69, 7%). Interestingly, their results were similar to ours, showing that 80% of lung cancer diagnoses happened after IPMN diagnosis, whereas none of colon cancer cases occurred after IPMN diagnosis [4]. Another study conducted by Osanai et al., who included 148 patients, with 35 patients having an EPM, found that most EPM cases were colon cancer (N = 11), followed by gastric cancer (N = 8) and lung cancer (N = 5). Overall, 80% (N = 4/5) of lung cancer cases occurred after IPMN diagnosis, but only 18% of colon cancer (N = 2/11) and 25% of gastric cancer (N = 2/8) cases occurred after IPMN diagnosis. This distribution is consistent with our study [19]. Consequently, we observed that lung cancer is prevalent in Eastern countries, with most cases occurring after IPMN diagnosis. This finding suggests a valuable target for follow-up. Another notable finding was the occurrence of hepatocellular carcinoma in our study, which was less common in other studies, regardless of whether conducted in Eastern or Western countries. Notably, 55% (N = 5/9) of these cases occurred after IPMN diagnosis. The high prevalence of chronic hepatitis B virus infection in Taiwan may be a contributing factor.

The mechanism of EPM occurrence is still not clearly understood. One possible hypothesis is that EPMs and IPMNs might share the same gene mutations, such as the k-ras mutation, which is associated with a variety of highly fatal cancers, including pancreatic ductal adenocarcinoma (PDAC), non-small cell lung cancer (NSCLC), and colorectal cancer (CRC) [23]. During the progression of IPMN to IPMC, the accumulation of several mutated genes was reported previously. House et al. found that IPMNs with invasive characteristics often exhibit multiple methylated genes. These genes are related to cell cycle control (p16, p73, and APC), DNA repair (MGMT and hMLH1), and cell adhesion (E-cadherin) [24]. Biankin et al. reported a higher frequency of loss of p16INK4A and Smad4, cyclin D1 overexpression, and p53 accumulation in IPMC, especially when associated with invasive carcinoma [25]. Some of these gene mutations were associcated with colon cancer and lung cancer as well [26,27,28]. In addition, the DNA damage checkpoint pathway, including the ATM-Chk2-p53 pathway, is involved in preventing the progression of several tumors, including colon cancer, lung cancer, and bladder cancer. In contrast, disturbance of this pathway due to Chk2 inactivation or p53 mutation contributes to the carcinogenesis of these cancers, as well as IPMN [29].

Few risk factors have been found to be associated with EPM occurrence. Controversial results regarding old age have been reported in previous studies [10,12,17]. One retrospective study demonstrated that IPMN patients with an EPM had higher rates of relatives with colorectal cancer (31%). Two of the fifty-one genetically tested patients (4%) were BRCA2 mutation carriers, and both had first-degree relatives with pancreatic cancer. This implies the genetic association between IPMN and EPMs [20]. Moreover, mounting evidence indicates that systemic inflammation activated by cancer cells accelerates tumor progression by stimulating cancer cell proliferation and metastasis, as well as by promoting angiogenesis and repairing DNA damage [30]. Several systemic inflammatory biomarkers have been examined to predict prognosis in different types of cancer, including C-reactive protein (CRP), neutrophil-to-lymphocyte ratio (NLR), platelet-to-lymphocyte ratio (PLR), and modified Glasgow Prognostic Score [31,32,33]. Given the association between these systemic inflammatory biomarkers and malignancies, we attempted to evaluate whether NLR or PLR could be utilized as a predictive factor for EPM occurrence. Unfortunately, our results did not show a significant association between either of these biomarkers and EPM occurrence.

However, we discovered that cyst size progression is not only an independent risk factor for the occurrence of EPM but also a predictive factor after the diagnosis of IPMN. This analysis demonstrated a very high test quality, which has not been reported in previous studies. For clinical utilization, we further identified that a cut-off value of a size increase of more than 1 cm had high accuracy (79%) and sensitivity (88%). Increasing evidence has proven the association between cyst size and the malignant transformation of IPMN. Based on previous studies, cyst size and the rate of cyst progression are associated with pancreatic malignant potential. Sadakari, Y., et al. concluded that an IPMN size of 30 mm or more tended to be associated with pancreatic malignancy, especially combined with an MPD of 5 mm or more [34]. A retrospective study that enrolled 189 IPMN patients with a median follow-up time of 56 months (range: 12–163 months) demonstrated that patients developing worrisome features had greater rates of BD-IPMN growth (2.84 mm/year vs. 0.23 mm/year; *p* < 0.001). The odds ratio of developing worrisome features increased with each unit (mm) increase in cyst size (odds ratio, 1.149; 95% CI, 1.035–1.276; *p* = 0.009). They concluded that low-risk BD-IPMN might require close surveillance if the cyst size increases by more than 2.5 mm/year [35]. Another retrospective study included 284 IPMN patients undergoing surveillance for a median follow-up duration of 56 months and reported that a faster growth rate was seen in cysts that developed into malignant IPMNs compared to benign cases (18.6 vs. 0.8 mm/year, *p* = 0.05) [36]. While increasing data have proved that cyst size progression implies malignant transformation in IPMN, our results also indicate an association with EPM occurrence. However, the mechanism of this finding needs to be studied in the future, and we believe that gene mutation and the immune microenvironment might play important roles.

There are some limitations to our study. First, this is a retrospective and single-center study with a limited number of patients. However, it is the first study in Taiwan with long-term follow-up compared to previous studies, thus providing some critical information. Secondly, the IPMN diagnosis in our study was only based on MRI diagnosis, without cystic fluid analysis. Most pancreatic cysts with serial image follow-up are classified as low-risk IPMN; thus, there is no indication for aspiration and fluid analysis, according to current guidelines [37,38].

## 5. Conclusions

In summary, we conducted the first study in Taiwan with a median follow-up duration of ten years and demonstrated the distribution of EPMs in IPMN patients. Our study demonstrated that over half of malignancies occurred after IPMN diagnosis. In addition to colon cancer, lung cancer also occurred frequently, especially after IPMN diagnosis. Cyst size progression is an independent risk factor of EPMs and a predictive factor of EPM occurrence after IPMN diagnosis according to our results, which was not found in previous studies. However, the pathophysiology needs to be clarified further. Furthermore, we suggest considering colonoscopy or immune fecal occult blood test, chest X-ray/low-dose lung CT, and abdominal ultrasound for cancer screening, especially when the size of the IPMN cyst progresses more than 1 cm, due to the higher risk of EPMs reported in our study.

## Figures and Tables

**Figure 1 cancers-16-04102-f001:**
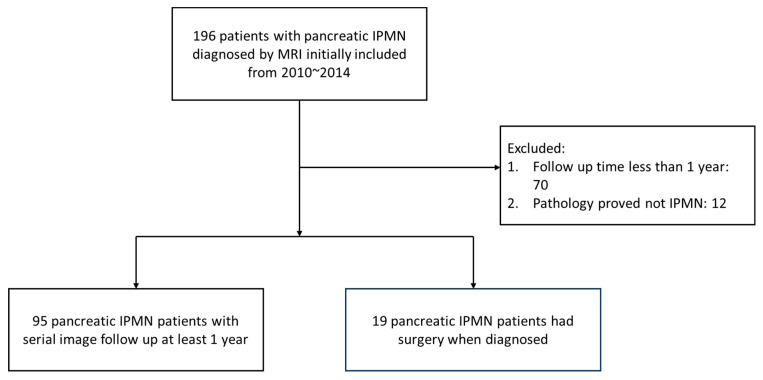
The flowchart of patients assessed.

**Figure 2 cancers-16-04102-f002:**
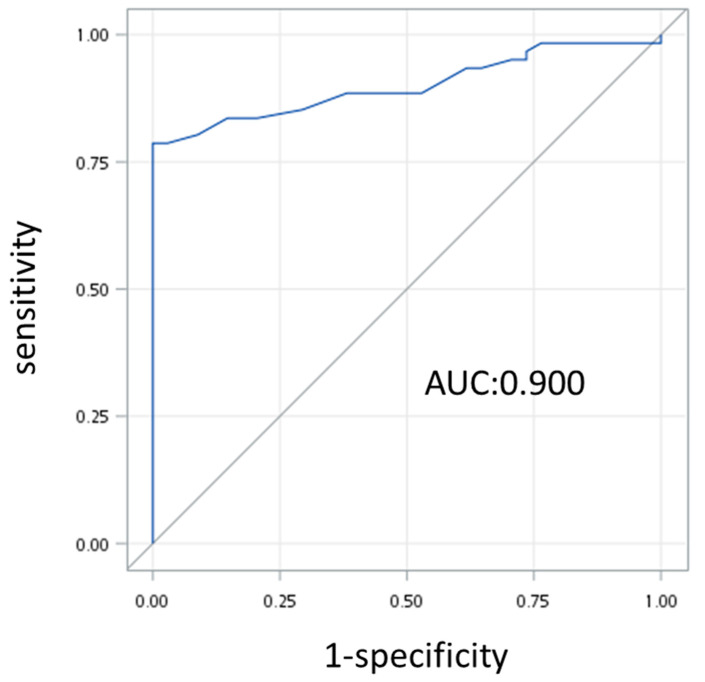
AUROC curve demonstrating the ability of cyst size progression to predict EPM occurrence in our IPMN patients.

**Figure 3 cancers-16-04102-f003:**
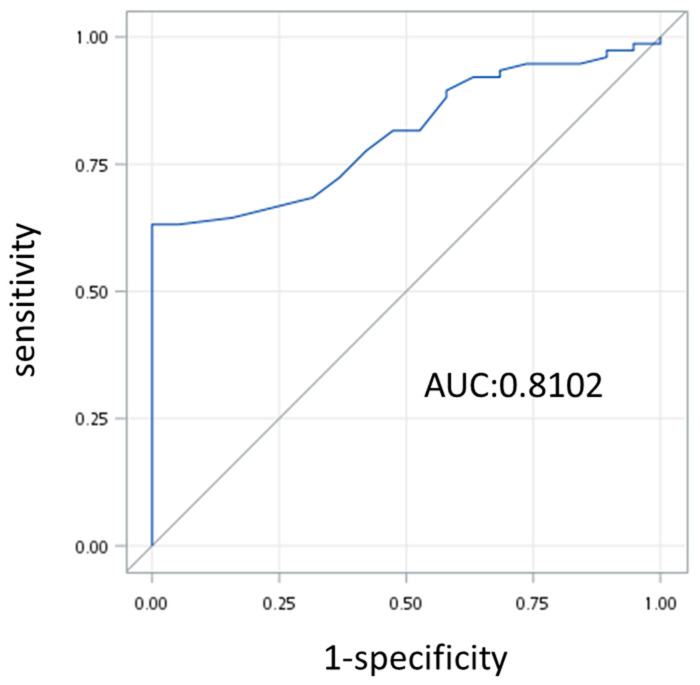
AUROC curve demonstrating the ability of cyst size progression to predict EPM occurrence only after IPMN diagnosis.

**Table 1 cancers-16-04102-t001:** Patient characteristics.

	Total (N = 114)	With EPM (N = 42)	Without EPM (N = 72)	*p*-Value
Age				0.020
<65 y/o	52 (45.6%)	14 (33.3%)	38 (52.8%)	
≥65 y/o	62 (54.4%)	28 (66.7%)	34 (47.2%)	
Gender				0.114
Male	57 (50.0%)	23 (54.8%)	34 (47.2%)	
Female	57 (50.0%)	19 (45.2%)	38 (52.8%)	
IPMN type				
Main duct type	10 (8.7%)	3 (7.1%)	7 (9.7%)	0.249
Branch duct type	102 (89.5%)	39 (92.9%)	63 (87.5%)	0.177
Mixed type	2 (1.7%)	0 (0.0%)	2 (2.8%)	0.397
Location				
Uncinate process	39 (34.2%)	16 (38.1%)	23 (31.9%)	0.129
Head	25 (21.9%)	9 (21.4%)	16 (22.2%)	0.185
Neck	17 (14.9%)	7 (16.7%)	10 (13.9%)	0.195
Body	58 (50.9%)	20 (47.6%)	38 (52.8%)	0.134
Tail	17 (14.9%)	7 (16.6%)	10 (13.9%)	0.195
Multifocal	31 (27.2%)	13 (30.9%)	18 (25.0%)	0.135
Cyst number				0.199
<3 cysts	107 (84.2%)	35 (83.3%)	59 (81.9%)	
≥3 cysts	20 (15.8%)	7 (16.7%)	13 (18.1%)	
Cyst size (cm)	1.5 (1–2.2)	1.7 (1.1–2.4)	1.4 (0.9–2.2)	0.168
CBD size (cm)	0.45 (0.4–0.5)	0.5 (0.4–0.6)	0.4 (0.4–0.5)	0.277
P-duct size (cm)	0.2 (0.2–0.3)	0.2 (0.2–0.4)	0.2 (0.2–0.3)	0.943
Mural nodule	19 (16.7%)	11 (26.2%)	8 (11.1%)	0.025
Diabetes mellitus	35 (30.7%)	13 (30.9%)	22 (30.6%)	0.166
Lab data				
HbA1C (%)	6.55 (5.7–7.35)	6.25 (5.65–7.1)	6.8 (5.7–7.9)	0.257
AST (U/L)	20 (16–25)	22 (17–39)	19 (15–24)	0.028
ALT (U/L)	19 (15–25)	20 (16–40)	19 (14–25)	0.110
CEA (ng/mL)	1.75 (1.18–3.22)	2.26 (1.18–3.89)	1.64 (1.1–2.5)	0.087
CA-199 (U/mL)	14.59 (6.92–30.00)	20 (7.51–43.43)	13.5 (6.89–26.9)	0.200
Follow-up time (year)	10.45 (7.22–12.30)	10.12 (6.31–11.87)	10.7 (7.2–14.3)	0.343

Red color for *p* value < 0.05.

**Table 2 cancers-16-04102-t002:** Extra-pancreatic malignancies occurring before, concurrently with, or after the diagnosis of IPMN.

	Total	Diagnosed Before IPMN	Diagnosed with IPMN	Diagnosed After IPMN
Extra-pancreatic malignancy	47	11	10	26
Colon cancer	10 (21.3%)	3 (27.3%)	4 (40.0%)	3 (11.5%)
Lung cancer	10 (21.3%)	0 (0.0%)	2 (20.0%)	8 (30.8%)
Hepatocellular carcinoma	9 (19.1%)	3 (27.3%)	1 (10.0%)	5 (19.2%)
Urothelial cancer	5 (10.6%)	0 (0.0%)	2 (20.0%)	3 (11.5%)
Breast cancer	2 (4.3%)	1 (9.1%)	0 (0.0%)	1 (3.8%)
Ovarian cancer	2 (4.3%)	1 (9.1%)	0 (0.0%)	1 (3.8%)
Head and neck cancer	3 (6.4%)	2 (18.2%)	0 (0.0%)	1 (3.8%)
Bile duct cancer	1 (2.1%)	0 (0.0%)	1 (10.0%)	0 (0.0%)
Skin cancer	2 (4.3%)	0 (0.0%)	0 (0.0%)	2 (7.6%)
Prostate cancer	1 (2.1%)	1 (9.1%)	0 (0.0%)	0 (0.0%)
Kaposi’s sarcoma	1 (2.1%)	0 (0.0%)	0 (0.0%)	1 (3.8%)
Lymphoma	1 (2.1%)	0 (0.0%)	0 (0.0%)	1 (3.8%)

**Table 3 cancers-16-04102-t003:** Risk factors of EPMs in patients with serial image follow-up.

	Overall	With EPM	Without EPM	Univariate		Multivariate	
	(N = 95)	(N = 34)	(N = 61)	OR (95% CI)	*p*-Value	OR (95% CI)	*p*-Value
Age					0.110		0.817
<65 y/o	44 (46.3%)	12 (35.3%)	32 (52.5%)	Reference		Reference	
≥65 y/o	51 (53.7%)	22 (64.7%)	29 (47.5%)	2.023 (0.852–4.801)		1.151 (0.347–3.819)	
Gender							
Male	43 (45.3)	18 (52.9%)	25 (41.0%)	1.620 (0.696–3.771)	0.263		
Female	52 (54.7)	16 (47.1%)	36 (59.0%)	Reference			
IPMN type					0.461		
Main duct type	5 (5.3%)	1 (2.9%)	4 (6.6%)	0.432 (0.046–4.027)			
Branch duct type	90 (94.7%)	33 (97.1%)	57 (93.4%)	Reference			
Location							
Uncinate process	31 (32.6%)	13 (38.2%)	18 (29.5%)	1.479 (0.611–3.579)	0.385		
Head	19 (20.0%)	7 (20.6%)	12 (19.7%)	1.059 (0.373–3.007)	0.914		
Neck	16 (16.8%)	6 (17.6%)	10 (16.4%)	1.093 (0.359–3.323)	0.875		
Body	52 (54.7%)	17 (50%)	35 (57.4%)	0.743 (0.320–1.725)	0.489		
Tail	15 (15.8%)	6 (17.6%)	9 (14.8%)	1.238 (0.400–3.835)	0.711		
Multifocal	28 (29.4%)	12 (35.3%)	16 (26.2%)	1.534 (0.620–3.795)	0.354		
Cyst number					0.669		
<3 cysts	76 (80.0%)	28 (82.3%)	48 (78.7%)	Reference			
≥3 cysts	19 (20.0%)	6 (17.6%)	13 (21.3%)	0.791 (0.270–2.315)			
Cyst size (cm)	1.4 (0.9–2)	1.5 (1–2)	1.4 (0.8–2)	1.441 (0.935–2.221)	0.098	1.173 (0.590–2.332)	0.648
Cyst size progress (cm)				9.429 (3.565–24.942)	0.000	8.542 (1.979–36.862)	0.004
Cyst size progress rate (cm/year)	0.01 (0.0–0.1)	0.1 (0.1–0.3)	0.0 (0.0–0.0)	188.399 (5.08–999)	0.004	0.086 (0.001–10.220)	0.313
P-duct size (cm)	0.2 (0.2–0.3)	0.2 (0.2–0.3)	0.2 (0.2–0.3)	2.481 (0.269–22.879)	0.422		
Mural nodule	7 (7.37%)	5 (14.7%)	2 (3.3%)	5.083 (0.930–27.793)	0.060	3.568 (0.436–29.192)	0.235
Lab data							
CEA (ng/mL)	1.7 (1.1–3.0)	2.5 (1.2–4.0)	1.6 (0.9–2.1)	1.366 (1.030–1.811)	0.030	1.197 (0.843–1.701)	0.314
CA-199 (U/mL)	11.9 (6.8–42.6)	11.6 (7.2–27.3)	12.0 (5.5–23.4)	1.001 (0.983–1.020)	0.881		
NLR	2.8 (1.9–4.5)	3.1 (1.9–4.8)	2.5 (1.6–4.1)	1.054 (0.939–1.184)	0.373		
PLR	134.0 (104.7–180.6)	129.7 (92.3–195.7)	138.0 (105.7–170.9)	0.999 (0.994–1.004)	0.660		
HbA1C (%)	6.3 (5.7–7.3)	6.4 (5.7–7.1)	6.2 (5.7–7.4)	0.943 (0.655–1.359)	0.754		

Red color numbers mean *p* value < 0.05, indicate significant difference.

**Table 4 cancers-16-04102-t004:** Predictors of EPM occurrence after IPMN diagnosis in patients with serial image follow-up.

	With EPM	Without EPM	Univariate		Multivariate	
	(N = 19)	(N = 76)	OR (95% CI)	*p*-Value	OR (95% CI)	*p*-Value
Age				0.357		0.990
<65 y/o	7 (36.8%)	37 (48.7%)	Reference		Reference	
≥65 y/o	12 (63.2%)	39 (51.3%)	1.626 (0.578–4.578)		1.007 (0.318–3.191)	
Gender				0.836		
Male	9 (47.4%)	34 (44.7%)	1.112 (0.406–3.045)			
Female	10 (52.6%)	42 (55.3%)	Reference			
IPMN type				0.977		
Main duct type	0 (0.0%)	5 (6.6%)	0.001 (0.001–999)			
Branch duct type	19 (100.0%)	71 (93.4%)	Reference			
Location						
Uncinate process	8 (42.1%)	23 (30.3%)	1.676 (0.596–4.713)	0.327		
Head	4 (21.1%)	15 (19.7%)	1.084 (0.314–3.744)	0.898		
Neck	5 (26.3%)	11 (14.5%)	2.110 (0.633–7.039)	0.224		
Body	9 (47.4%)	43 (56.6%)	0.691 (0.252–1.893)	0.472		
Tail	3 (15.8%)	12 (15.8%)	1.000 (0.252–3.970)	1.000		
Multifocal	7 (36.8%)	21 (27.6%)	1.528 (0.530–4.406)	0.432		
Cyst number				0.898		
<3 cysts	15 (79.0%)	61 (80.3%)	Reference			
≥3 cysts	4 (21.0%)	15 (19.7%)	1.084 (0.314–3.744)			
Cyst size (cm)	1.7 (1.2–2.0)	1.3 (0.8–2.0)	1.266 (0.785–2.040)	0.333		
Cyst size progress (cm)			3.026 (1.538–5.952)	0.001	2.911 (1.446–5.861)	0.002
Cyst size progress rate (cm/year)	0.1 (0.1–0.3)	0.0 (0.0–0.1)	1.024 (0.567–1.849)	0.937		
P-duct size (cm)	0.2 (0.2–0.4)	0.2 (0.2–0.3)	2.921 (0.251–34.031)	0.392		
Mural nodule	3 (15.8%)	4 (5.3%)	3.375 (0.687–16.582)	0.099	2.502 (0.459–13.629)	0.288
Lab data						
CEA (ng/mL)	3.4 ± 2.4	1.6 (0.9–2.5)	1.059 (0.929–1.206)	0.390		
CA-199 (U/mL)	15.6 (6.6–29.1)	11.9 (6.8–24.3)	1.005 (0.986–1.025)	0.613		
NLR	3.1 (1.9–7.1)	2.7 (1.7–4.5)	1.009 (0.888–1.147)	0.889		
PLR	120.5 (90.0–171.0)	143.5 (105.7–195.7)	0.995 (0.987–1.003)	0.246		
HbA1C (%)	6.4 (5.7–7.1)	6.3 (5.7–7.4)	1.057 (0.710–1.574)	0.783		

Red color numbers mean *p* value < 0.05, indicate significant difference.

## Data Availability

Data are contained within the article.

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
