# Peer review of "The Distribution and Predictive Factor of Extra-Pancreatic Malignancy Occurrence in Patients with Pancreatic Intraductal Papillary Mucinous Neoplasm—A Ten-Year Follow-Up Case–Control Study in Taiwan"

_cancers, 2024, doi:10.3390/cancers16234102_

Round 1

Reviewer 1 Report

Comments and Suggestions for Authors

The study is interesting, authors present a long term retrospective study on more than a hundred patients with pancreatic IPMN to look for extrapancreatic malignancy. They identify most EPMs and new risk factors( progression of size )

In my opinion authors should try to complete the discussion and try to explainc, why do the other malignancies occur in patients with IPMN

Should the results of the study be the recommendation to perform colonoscpy?chest CT/X-ray?Abdominal US/CT?

I found mistake that makes the abstract not clear

"However, rare reports discussed EPM occurrence after EPM diagnosis and no risk factor is clearly" . It should be, I suppose , adter IPMN 

Reviewer 2 Report

Comments and Suggestions for Authors

Summary

In this retrospective study the authors evaluated the incidence of extra pancreatic malignancies in patients with IPMNs and investigated the risk factors associated with EPMs. At an average follow up of 10.45 years, 47 EPMs occurred in 42 patients (36.8%), either before, concurrently or after the diagnosis of IPMN. The most common EPM was colon cancer and lung cancer. Cyst size of greater than or equal to 1 cm was associated with a significant risk of malignancy.

Strengths

Interesting questions being addressed in this study.

Weaknesses

Poor study design.

Comments

1.       Most patients with EPMs were incidentally detected, how many patients had EPM as an indication for the imaging which picked up the pancreas cyst?

2.       Can you please add the indications for the original imaging which picked up the pancreas cyst for all the cases?

4.       How many of theseEPMs were picked up by the surveillance scans?

6.       What was the median age of the patient with EPMs diagnosed after IPMN diagnosis?

7.       In which location of the pancreas was the PD size measured?

8.       Can you please provide the incidence of EPMs in the general population in Taiwan and compare the IPMN population for the cancers that occurred after their diagnosis?

9.       This is a retrospective study with obvious selection bias, for instance only those patients with issues stayed in the system to get surveillance scans and probably the reason they were followed for close to 10 years, do you think we can extrapolate this information to the whole population?

Comments on the Quality of English Language

Can be improved.

Round 2

Reviewer 2 Report

Comments and Suggestions for Authors

Summary

In this retrospective study the authors evaluated the incidence of extra pancreatic malignancies in patients with IPMNs and investigated the risk factors associated with EPMs. At an average follow up of 10.45 years, 47 EPMs occurred in 42 patients (36.8%), either before, concurrently or after the diagnosis of IPMN. The most common EPM was colon cancer and lung cancer. Cyst size of greater than or equal to 1 cm was associated with a significant risk of malignancy.

Strengths

Interesting questions being addressed in this study.

Weaknesses

Retrospective nature of the study with obvious selection bias.

Comments

1.       What type of malignancies occurred before, concurrent and after IPMN diagnosis?

Author Response

Pleas see the attachment
